# Human Augmentation Technologies for Employee Well-Being: A Research and Development Agenda

**DOI:** 10.3390/ijerph19031195

**Published:** 2022-01-21

**Authors:** Bach Q. Ho, Mai Otsuki, Yusuke Kishita, Maiko Kobayakawa, Kentaro Watanabe

**Affiliations:** 1School of Engineering, Tokyo Institute of Technology, 2-12-1, Ookayama, Meguro-ku, Tokyo 152-8550, Japan; 2National Institute of Advanced Industrial Science and Technology, c/o Kashiwa II Campus, University of Tokyo, 6-2-3, Kashiwanoha, Kashiwa, Chiba 277-0882, Japan; mai.otsuki@aist.go.jp (M.O.); kentaro.watanabe@aist.go.jp (K.W.); 3School of Engineering, The University of Tokyo, 7-3-1, Hongo, Bunkyo-ku, Tokyo 113-8656, Japan; kishita@pe.t.u-tokyo.ac.jp; 4Chiba Institute of Technology, 2-17-1, Tsudanuma, Narashino, Chiba 275-0016, Japan; kobayakawa.maiko@p.chibakoudai.jp

**Keywords:** human augmentation, well-being, research and development, COVID-19, employee, grounded theory, job satisfaction, responsible research and innovation, teleworking, virtual reality

## Abstract

The COVID-19 pandemic has changed the style of work. In adapting to the changing work environment, human augmentation technologies (HAT) can provide employees with new options to support their work. However, the agenda for research and development of HAT for the new normal is still unclear. In this study, we set two research questions: (i) what type of technology demand has emerged among employees due to the COVID-19 pandemic; and (ii) what is the nature of job satisfaction experienced by employees during the COVID-19 pandemic? This study aims to clarify the technology demand and job satisfaction of employees during the COVID-19 pandemic. We analyzed data from in-depth interviews with employees based on a grounded theory approach to answer the research questions and proposed an agenda for the research and development of HAT to enhance employees’ well-being in this new normal based on the crosspoint of technology demand and job satisfaction. The theoretical contribution of this study is the development of models of technology demand and job satisfaction of employees during the COVID-19 pandemic. The practical contribution is the development of a crosspoint framework to enable the development of HAT to support work while considering their impact on employees’ well-being.

## 1. Introduction

COVID-19 has spread around the world since 2020 and has widened the physical and mental distance among people and changed the nature of work. Occupations that necessitate proximate contact with people perpetuate the risk of infection despite infection prevention efforts, while occupations that promote work from home force employees to undergo rapid changes in their work styles [1,2,3]. These enormous risks and changes in the work environment during the COVID-19 pandemic have increased the physical and mental burden on employees and reduced their well-being [4,5].

As the influence of COVID-19 grew stronger, the degree to which technology was introduced into the workplace increased. For example, the Global Workplace Analytics reported that 82% of U.S. employees wanted to work remotely at least once a week and 56% had a job where at least some of the tasks could be done remotely in 2020 [6]. Advanced technology can help adapting to the changing work environment in the new normal, i.e., the new state of society [7]. In particular, human augmentation technologies (HAT) can provide employees with new options to support their work in situations that require social distancing. HAT refer to technologies improving human functions by facilitating achievements beyond their capabilities [8]. However, technologies are not always developed for enhancing human well-being though HAT have potential to improve the workplace environment of employees [9]. Advanced technologies may harm employee well-being while improving the efficiency of work. Therefore, research and development (R&D) of HAT for the new normal in workplaces is needed.

Employee well-being has been considered with frameworks assessing both positively and negatively affecting elements in management studies [10,11]. To enhance employee well-being based on HAT, this study focuses on technology demand addressing insufficient technological resources and job satisfaction positively affecting employee well-being. Consequently, we explore the following two research questions (RQ).
RQ 1: What type of technology demand has emerged among employees due to the COVID-19 pandemic?RQ 2: What is the nature of job satisfaction that employees experience during the COVID-19 pandemic?

This research aims to clarify the technology demand and job satisfaction of employees (including self-employed workers) during the COVID-19 pandemic and to provide the model describing future directions for R&D of HAT. The remainder of this paper is structured as follows. First, the grounded theory approach employed in this study is explained in Section 2. Next, the answers to the two RQs are presented in Section 3, and an agenda for R&D of HAT is developed as a crosspoint framework based on the technology demand and job satisfaction in Section 4. The contributions of this research and future research directions are discussed in Section 5.

## 2. Materials and Methods

A grounded theory approach (GTA) based on in-depth interviews was adopted to understand the situation and sentiment of employees during the COVID-19 pandemic. GTA is a text analysis method based on pragmatism that aims to construct an understandable theory by abstracting complex phenomena of human interaction, such as work operations [12]. To answer the RQs, we asked the following questions: “What type of demand about HAT has emerged in work operations after the COVID-19 pandemic?” and “What is your job satisfaction after the COVID-19 pandemic?” The text data were analyzed using MAXQDA 2020.

The data analysis was based on GTA and involved three coding steps: open coding, axial coding, and selective coding [13,14,15]. In open coding, we generated categories for all collected data by assigning labels to each chunk of text to briefly describe the meaning of the text. In axial coding, similar categories were aggregated and organized hierarchically. In selective coding, a theory describing the phenomenon was constructed based on the relationships among the categories generated in the axial coding.

The respondents were employees interested in HAT, selected using snowball sampling. Snowball sampling is a sampling technique in which survey respondents with specific characteristics and knowledge in line with the research topic are approached through the introduction of existing respondents [16]. We also considered the balance of gender, age, and occupation when recruiting respondents. Finally, theoretical saturation, which refers to the state in which no new categories are generated for a theory even when new data are added [17], was reached for both RQ1 and RQ2 with 20 respondents. The profiles of the 20 respondents are shown in Table 1. The in-depth interviews were conducted using the video call tools Zoom or Microsoft Teams, and each session took about 1 h on average. All interviews were recorded and transcribed for analysis.

## 3. Results

### 3.1. Technology Demand

By using GTA to analyze the textual data from the interviews, we found that the technology demands of employees in the COVID-19 pandemic can be categorized into three categories: “overcome access barriers,” “beyond reality,” and “pursue reality.” These three categories represent one demand to overcome barriers to access technology (“overcome access barriers”) and two demands, based on the reality axis, to achieve unrealistic functions using technology (“beyond reality”) or feel the same perception as in reality (“pursue reality”) (Figure 1).

#### 3.1.1. Overcome Access Barriers

“Overcome access barriers” refers to the demand of overcoming barriers in access to technology. HAT hold the potential to significantly improve the work environment, but they also cause barriers to access. This category was composed of two subcategories: “ease of use” and “resolving hardware limitations”.

“Ease of use” refers to the demand of using a new technology with minor learning cost. Typical factors that reduce the ease of use are the difficulty in interface design and usability. In addition, depending on an individual employee’s level of technology readiness [18] and the differences in cultural customs surrounding them, there may be significant learning costs to access HAT. While employees want to use new technologies with low learning costs, ease of use does not always result in a positive outcome. Excessive ease of use can lead to negative social changes (Respondent 19). Some sample responses are presented below.


*It is great to have new technology, but in the end, the first tutorials need to be taught intensively to the students. Other than watching videos, typing and writing reports is a bit difficult if they have not been trained before.*

*(Respondent 10)*



*An old lady called me, and I asked her if she would like to join a course I am conducting over Zoom. However, although I suggest it, she cannot do it. I think if there was a very simple system, it would make communication much easier.*

*(Respondent 20)*



*With the advancement of remote technology, the hurdles to buying plants are being lowered that is excellent. However, plants are essentially life, and in the sense of buying life, I think it is a very serious matter. I am afraid that with the advancement of remote technology, people will not recognize plants as living beings.*

*(Respondent 19)*


“Resolving hardware limitations” refers to the demand for solutions to the limitations of the physical environment and hardware when using technology. Although work from home provides employees a certain amount of freedom, it requires them to use technologies including HAT in less-equipped environment such as their home, unlike workplaces provided by companies. Therefore, how engaged they are in the technology and how they set up their own environment determine whether they can fully access the functions of HAT. Furthermore, there are individual differences in literacy in deciding what type of environment should be set up. Sample responses are presented below.


*After conducting the online class, I thought it would be nice if I could make handwriting easier. Some of the students took pictures and uploaded their paper notes because it was very difficult to write notes on a smartphone. However, even if you take a picture and upload it, you cannot search for text in that picture. I wonder if there is anything we can do about this.*

*(Respondent 10)*



*I used to work on a desktop PC with a big screen at the workplace, but when I started teleworking and used a laptop with a small screen, I found it difficult to work. The monitor is small and the keyboard is also small, so my shoulders become stiff and my work efficiency deteriorates. *

*(Respondent 18)*


#### 3.1.2. Beyond Reality

“Beyond reality” refers to the demand for technology to perform functions that could not be achieved in reality, and thus far, most of the demand for HAT has been regarding this. This category consisted of four subcategories: spatial augmentation, time augmentation, capability augmentation, and knowledge augmentation.

“Spatial augmentation” refers to the demand for connecting and interacting with people geographically distant from each other and for virtual expansion of physical space that is limited in reality. The demand for remote control of robots and machines using HAT and remote interaction with others is typical of the demand for spatial augmentation, as the research on this technology has been aimed at so far. In addition, a demand was revealed for the use of HAT to allow an unrealistic number of people to visit places they cannot be admitted to in reality (Respondent 6). In other words, the employees wanted virtual and physical augmentation of a specific space in reality. Sample responses are presented below.


*There is a shortage of human resources in rural areas, particularly young and talented people who have left for cities. Therefore, local people are looking for ways to allow talented people to access local industries through virtual spaces, or to participate in project-based activities.*

*(Respondent 5)*



*A tourist agency provides an online tour of the restricted area where only scholars are usually allowed to enter. Then, they provide a Zoom tour for 600 people that is impossible to do offline. They provide products that only online tours can do.*

*(Respondent 6)*


“Time augmentation” refers to the demand for a virtual flow of time different from reality. A typical demand in this subcategory is that to be able to communicate smoothly with others without synchronizing time. Text communication over the Internet has been used for this type of demand. However, the existing chat tools handle ‘stock’ information stored for a long period and ‘flow’ information used in an ad-hoc manner, separately. The demand for supporting employees’ information management was mentioned in the interviews (Respondent 5). In addition, there was a demand for the possibility of experiences in a short time that would take a long time in reality (Respondent 6). Sample responses are presented below.


*Conversations over the chat tools in our company flow so fast that it is sometimes difficult to find important information or it is likely to be overlooked, so I am still looking for some technologies to overcome this problem.*

*(Respondent 5)*



*For example, you can plan a tour of two distant places in one hour in online tours. Also, seasons. You can view the cherry blossoms, the autumn leaves, and skiing.*

*(Respondent 6)*


“Capability augmentation” refers to the demand for augmenting physical and cognitive capacity through the use of HAT. Employees expected not only the augmentation of their capabilities, but also the improvement of their work skills and health as outcomes of the augmentation. In addition, there was a demand for technology to enhance their looks (Respondent 17). Two sample responses are presented below.


*It is called the cocktail party effect, that is, there is a lot of information in our brains, but we only take the information we want. The brain of a child with disability cannot do this. They pick up all the sounds, even which they do not need. Elderly people also tend to pay attention to information that they do not need. I think it would be great if we could use virtual reality devices to effectively limit their attention.*

*(Respondent 3)*



*There are video calls that transform your face into a specific character. When adults talk on the video call with their children, the children get bored and sometimes even shift into a bad mood; so by letting them play with augmented reality, the adults can have a calm conversation.*

*(Respondent 17)*


“Knowledge augmentation” refers to the demand for gaining expertise to improve one’s judgment and thinking. The borrowing of expertise is not limited to humans, but can also be aided by artificial intelligence (AI). By receiving feedback from HAT that employees are unaware of, they can improve their work skills. The sample response is presented below.


*I wish there were sensors attached to the body that would sound an alarm to tell the user that the angle of the arm is decreasing or that it should be straighter. It would be useful if there would be a system that displays helpful information on the monitor screen, displays the progress, and provides an evaluation.*

*(Respondent 3)*


#### 3.1.3. Pursue Reality

“Pursue reality” is the demand to be able to perform and perceive in the same manner as in reality, even when using HAT. From the analysis results, the demand for HAT is not only for the realization of unrealistic functions, but also for the restoration of reality through technology. This category is composed of three subcategories: realism, others’ attention, and connectedness.

“Realism” refers to the demand for HAT to be able to achieve the same perceptions as in reality. In addition, there was a demand to be able to move around in the virtual space in the same manner as in reality. A sample response is presented below.


*It is very meaningful to visit a place where NPOs are actually working and to experience it with your senses. You can see how such a great person is working in such a small and empty office with such a massive goal, and how you are actually doing in contrast. You can compare their work style with yours. It would be worthwhile to be able to experience this in virtual reality.*

*(Respondent 4)*


“Others’ attention” refers to the demand to be able to perceive the presence of others as in reality using HAT [19,20] and to experience situations where others are watching users. As a result of the changes in the work environment during the COVID-19 pandemic, some employees felt lonely due to physical distance from others and were less able to concentrate on their work. To solve this problem, a situation in which employees can concentrate on their work with others’ attention must be constructed. There was also a demand for facilitating communication by perceiving the presence of others. A sample response is presented below.


*Ambient noise is quite important to me. I can work better in a place where there is a lot of noise than in a quiet place, and when I feel someone’s eyes around me, I feel I have to do my job properly.*

*(Respondent 10)*


“Connectedness” refers to the demand to communicate with others in the same manner as in reality, using HAT. While spatial augmentation based on technologies has made it possible to communicate remotely, it also requires a different manner of communication from face-to-face communication that has led to new detriments [21,22]. Employees demanded to overcome negative effects of remote communication and regain the feeling of face-to-face communication. Sample responses are presented below.


*It would be nice to be able to enter in a virtual reality and have more information than what is on this 2D screen, but that alone does not bring significant differences from the current online tour. The experience will change when mutual connections can be made in the virtual reality space, I mean, when customers can communicate with each other.*

*(Respondent 6)*



*When I am talking on a video call like this, even if I am talking to a group of people, the conversation seems often like the accumulation of one-on-one conversations, so I feel it is not suitable for lively conversations. Tools that can improve communication are necessary.*

*(Respondent 18)*


### 3.2. Job Satisfaction

The categories generated for job satisfaction by the GTA analysis converged on three motivation theories: self-determination theory, needs theory, and mastery theory. The results revealed that job satisfaction in the COVID-19 pandemic can be classified with two axes: locus of control [23] and intrinsic motivation [24] (Figure 2).

The self-determination theory explains that employees obtain job satisfaction derived from intrinsic motivation based on internal control. In other words, according to the self-determination theory, employees enhance their well-being when they obtain job satisfaction by fulfilling their internal needs for things they can control, such as recognition of their high skills in work. The needs theory explains job satisfaction derived from extrinsic motivation based on external control. When employees obtain job satisfaction from satisfied extrinsic needs for things out of their control, as in the case of increased wages according to better customer response, they increase their well-being, as per the needs theory. Finally, the mastery theory explains that job satisfaction comes from the source of intrinsic motivation based on external control. Accordingly, employees enhance their well-being when they obtain job satisfaction by satisfying their internal needs to overcome obstacles in the work environment that they cannot control.

#### 3.2.1. Self-Determination Theory

Self-determination theory describes the concept of intrinsic motivation that explains that people experience pleasure in the behavior itself, driven by their inner psychological needs without being influenced by external rewards [25]. Employees follow self-determination theory when they focus on job satisfaction generated by intrinsic motivation that is aroused by objects that they can control. Self-determination theory includes three psychological needs: autonomy, relatedness, and competence [26]. Our findings revealed that job satisfaction derived from intrinsic motivation based on internal control converged with these three needs.

Autonomy refers to the need to feel that one’s actions are autonomous. Employees obtain job satisfaction when they are responsible for their own work performance and when they can design their own work environment. The interviews revealed that the subcategory of autonomy comprised of the job satisfaction experienced by having sufficient self-discretion. Two sample responses are presented below.


*Travel agencies often have a division of labor, and in many cases, the person planning the tour and the person going to the site as a guide are different. As a sole proprietor, I go to the site to take pictures, think about how I would guide the tourists, and when I finally advertise for the tourists and they show up, I feel extremely happy. I realize that they have really come.*

*(Respondent 6)*



*I once got a job after I got my doctoral degree. I liked my company and thought I would work there for the rest of my life. However, I was transferred to a department of R&D in the company that communicated with other research institutes, and I was able to compare myself as a researcher in a company with people in research institutes. In the process of comparing the two, I came to realize that research institutes have an overwhelming advantage in terms of being able to research my own ideas on my own responsibility. In a company, you have to work within the framework of the company, for example, “That’s interesting, but it doesn’t fit into our business.” However, in a research institute, you can take responsibility for your own work, so I quit my job and moved to a research institute.*

*(Respondent 14)*


Relatedness refers to the need for psychological connection with others. The interviews revealed that employees are intrinsically motivated to interact with others and obtain job satisfaction through interactions, which is not limited to customers, but also includes a sense of solidarity as a group while working with colleagues. A sample response is presented below.


*There are things that I cannot do but others can. The process of complementing each other’s strengths and weaknesses to create something that I would never have been able to do on my own is interesting.*

*(Respondent 5)*


Competence refers to the need to demonstrate one’s superior ability and demonstrate high performance. Employees perceive a sense of competence and obtain job satisfaction by demonstrating their inherent abilities while immersing themselves in their work and increasing their confidence in their abilities. The interviews revealed that the subcategory of competence comprised of job satisfaction derived from the employees’ enthusiasm for their work driven by intrinsic motivation to demonstrate their abilities. A sample response is presented below.


*I feel that I am able to fulfill my role in the company because I am the only one who can do what no one else can.*

*(Respondent 1)*


#### 3.2.2. Needs Theory

Needs theory proposed by McClelland categorizes employees’ motivations into three categories: power, achievement, and affiliation [27]. Employees essentially follow the needs theory when they focus on job satisfaction derived from extrinsic motivation aroused by external factors that they cannot control, such as other people (e.g., colleagues, customers) or the surrounding environment. The results showed that job satisfaction derived from extrinsic motivation based on external control converged with the three motivations of the needs theory. These three motivations can be classified along two axes: goal attainment and attachment to a specific object [28].

Power motivation is the need for high goal attainment and attachment to the object, and represents the motivation to seek pleasure in one’s own influence on others. The interviews revealed that the subcategory of power motivation comprised of job satisfaction that arises from contributing to customers, the company, and society through work. The sample responses are presented below.


*It is great to be able to help someone, or to be able to play a role in empowering someone else in our own way.*

*(Respondent 4)*



*I had a lot of opportunities to consult with my parents about their families, so I was able to experience many things that allowed me to solve family problems for their children and lead to their future. In a way, I think it is very meaningful outside the classroom.*

*(Respondent 12)*


Achievement motivation is the motivation to prioritize the attainment of goals rather than attachment to a particular object. The interviews revealed that the subcategory of achievement motivation comprised of job satisfaction obtained by the sense of achievement through job performance. The difference between competence in self-determination theory and achievement motivation is that competence focuses on one’s perception (intrinsic motivation) of demonstrating one’s inherent ability (internal control), while achievement motivation focuses on extrinsic motivation that one cannot control, such as positive feedback from others. A sample response is presented below.


*When students thank me for teaching them how to pass examinations for qualifications, I feel that it is worthwhile and I am glad to have done it.*

*(Respondent 11)*


Affiliation motivation prioritizes attachment to an object over goal achievement. The interviews revealed that job satisfaction obtained through attachment to one’s work content formed this subcategory. Employees feel attached to their work and happy to be engrossed in their every day. Sample responses are presented below.


*My friend said, “My tours are like my own children.” The tour is like a living organism, improving and refining as it goes along, so I feel very attached to the tour that I organize myself, and it is very motivating and fun.*

*(Respondent 6)*



*I think plants are very interesting because they are simple. You just need to be barefoot and feel the soil in the field, you really do not need words or anything.*

*(Respondent 19)*


#### 3.2.3. Mastery Theory

Mastery theory describes the preference for challenging difficult goals in the external environment according to inner intellectual curiosity, and the need to become mature in relation to one’s surroundings through new discoveries [29,30]. Only the “challenge” subcategory is included in this category. The interviews revealed a convergence in this subcategory of job satisfaction with the challenge of new tasks and the discoveries acquired from the challenge. Sample responses are presented below.


*I handle data that the client could not handle, such as special questionnaires and detailed performance data. It is very motivating to report findings that can be applied to management. It is very challenging for me to work on something that is new to our company.*

*(Respondent 1)*



*Currently, high technology is being used in classes. In that sense, I think it is fun to teach in a new style.*

*(Respondent 10)*


## 4. Discussion

### 4.1. Crosspoint Framework for R&D Agenda of HAT

A framework consisting of nine domains was constructed from the crosspoint of technology demand and job satisfaction. Table 2 presents the examples of R&D topics to be promoted in each domain, but the topics to be addressed in each domain are not limited to the list in the table. The framework provides a comprehensive view of the R&D agenda for HAT that enhance the well-being of employees.

### 4.2. Overcome Access Barriers × Self-Determination Theory

The domain at the crosspoint of “overcome access barriers” and self-determination theory emphasizes the importance of satisfying intrinsic motivation while improving the employees’ access to technology. Employees can enhance their well-being by autonomously accessing technology.

*Improving interface usability*—User interfaces in virtual spaces have a significant impact on technology usability [31]. Low usability decreases the employees’ autonomy in using technology [32] and discourages their job satisfaction. Similarly, tasks that require remote operation of robots and machines are also prone to the problem of operational autonomy. However, if an employee has sufficient work experience, the learning cost may be reduced. If employees are familiar with HAT in their private life, they may become accustomed to operating the technologies more efficiently and promptly than the other employees. In contrast, complex jobs will require more learning costs than the simple ones. Pertaining to this topic, the following research questions need to be addressed: Which interfaces increase learning costs? How much do the employees’ work experience and technology literacy reduce the learning costs? How does job complexity increase the learning costs?

*Improving operation ability by perceptual stimulation*—It is useful to stimulate the employees’ perceptions such that they can access the functions of HAT while promoting their sense of competence. For example, stimulating the employees’ olfactory and auditory senses in the workplace can increase their concentration and enhance their ability to use HAT. In addition to smell and hearing, technologies have been developed to stimulate other senses as well [33,34]. The combination of these supplementary HAT could enhance the operational capability of the main HAT for performing tasks. However, the operational capabilities enhanced by perceptual stimuli in reality may not influence the virtual space. As work in virtual spaces increases, it will be necessary to develop technologies that stimulate perception in virtual spaces. Pertaining to this topic, the following research questions need to be addressed: How can main and supplementary HAT be combined to perform tasks? How can the stimulating effects of perception in reality be carried over to the virtual space? How can perceptions be stimulated in a virtual space?

### 4.3. Overcome Access Barriers × Needs Theory

The domain at the crosspoint of “overcome access barriers” and needs theory emphasizes the importance of enabling employees to access technology while achieving their goals or stimulating attachment to objects. Encouraging access to technology by promoting engagement with the surrounding environment and others can enhance the employees’ well-being.

*Imparting personality to instructional systems*—In the process of accessing HAT, employees can become attached to the technology they are using, thereby promoting job satisfaction through affiliation motivation. AI can be used to impart personality to instructional systems such as personal assistants in HAT [35]. Wizards, the assistant software within the technology, should not only be easily understandable, but also be easy to be intimate with. The possibility to change the wizard’s personality, looks, and voice according to the user’s preference will make employees enthusiastic about HAT, similar to how the character Theodore fell in love with Samantha when he recognized her personality in the movie *Her*. However, assigning beauty to every wizard is not always effective. Pertaining to this topic, the following research questions need to be addressed: How does a wizard’s looks and voice affect employee accessibility? Does changing the wizard’s personality to suit employee preferences contribute to accessibility? Which affinities exist between the function and form of technologies and the wizard’s personality, looks, and voice?

*Building an online community*—To stimulate the employees’ power motivation, it is effective to build a web platform that connects users of HAT globally and to develop an online community. Participants in the community can feel a sense of contribution by teaching each other how to use technologies [36], improving employees’ accessibility to the HAT and enhancing their well-being. Furthermore, introducing competitive mechanisms may lead to open innovation [37]. The widespread diffusion of upgraded HAT based on user ideas can also satisfy the power motivation of employees. Collective wisdom [38] can be leveraged to increase the accessibility of HAT across a broad spectrum of society. Pertaining to this topic, the following research questions need to be addressed. How can users be encouraged to interact with each other on a platform? How can the perceived sense of the contribution of individual users in the community be encouraged? How can the insights gained within the platform be effectively diffused to outside the platform?

### 4.4. Overcome Access Barriers × Mastery Theory

The domain of the crosspoint of “overcome access barriers” and mastery theory emphasizes the importance of facilitating access to technology while stimulating the employees’ challenges and curiosity. Employees can enhance their well-being when they simultaneously acquire new skills and access HAT.

*Stepwise unlocking of technological functions*—Gamification [39] can be applied to design a work system that allows employees to acquire skills in operating HAT, while enjoying the process of developing them further. Employees’ job satisfaction can be enhanced by increasing the range of available technology functions or additional attachments, as employees gain more expertise in HAT. However, it is arguable whether the criteria for employee mastery should be determined by the developers of the technology or by the individual company. We should also be cautious about how the technology functions should be unlocked. To maintain employees’ job satisfaction, it would be better to unlock the functions gradually. In addition, it is not always a single player who challenges in gamification [40]. Decisions about how much technology functions should be unlocked for each employee are more complex in the case of group work. Pertaining to this topic, the following research questions need to be addressed: How can the mastery level of employees be measured? How should mastery be related to the operational range of technologies? How should the unlocking of technology functions be determined in the case of group work?

*Restoring physical and cognitive functions and supporting working*—HAT can be used to restore the physical and cognitive functions of the elderly and the physically impaired people to assist them in finding job opportunities. Global demand for utilizing the labor force of the elderly is increasing as they account for a large proportion of the world population. Restoring their functions and supporting their employment will not only improve their well-being but also enhance the sustainability of the social system [41]. With the development of HAT, even bedridden people can now remotely operate avatar robots to serve customers [42]. It is unclear how much of a physical and mental burden the use of HAT will have on physically impaired people, but for now, the positive effects of reconnecting them with society through technologies and enabling them to engage in new challenges seem to be significant. Furthermore, HAT can enhance the well-being of physically impaired people by supporting their rehabilitation, in addition to assisting them to work [43]. As HAT continue to be developed for supporting the employment of the elderly and physically impaired people, social institutions will be improved. In the future, technology developers may also be required to be experts in the domain of social institutions. Pertaining to this topic, the following research questions need to be addressed: How can HAT contribute to restoring the functions of the lives of the elderly and the physically impaired people? How does the use of HAT burden the elderly and the physically impaired people? How can social institutions be designed to support the employment of the elderly and physically impaired people through HAT?

### 4.5. Beyond Reality × Self-Determination Theory

The domain of the crosspoint of beyond reality and self-determination theory emphasizes the importance of stimulating intrinsic motivation in augmenting the human functions of employees. Employees can enhance their well-being by being able to perform tasks that they could not do in reality, without being controlled by the surrounding environment or the technologies they use.

*Visualizing potential ability and conditions*—Visualization of the employees’ hidden abilities and conditions (e.g., health) is useful for encouraging self-determination. For example, by using sensing technologies to visualize the degree of fatigue, employees can self-adjust their workload. However, the visualized data must be handled carefully. Employees may receive unfairly low rewards if negative data are shared with the company [44]. Visualization techniques must also be carefully considered. Work will stagnate if employees always have to check the monitored information. Intuitive and understandable visualization should be made available to employees with minimal effort. Pertaining to this topic, the following research questions need to be addressed: What potential abilities and conditions should be visualized? Which visualized data should be shared with the company? What types of visualization techniques are useful?

*Moderating disparities in augmented capabilities*—When there are multiple employees using HAT in the same task, an unreasonable disparity in the augmented capabilities of each employee will reduce the sense of competence and decrease job satisfaction. However, simply adjusting the augmented capabilities of other employees to match that of the employees with the lowest augmented capabilities will only reduce the work efficiency. It is important to establish rules for technology usage that employees can accept. Changing the combination of the augmented capabilities by replacing the team members can also be an effective approach. Pertaining to this topic, the following research questions need to be addressed: What types of disparities do employees complain about? How can both the reduction of disparities and the improvement of operational efficiency be pursued? How should augmented capabilities be allocated within an organization?

### 4.6. Beyond Reality × Needs Theory

The domain of the crosspoint of beyond reality and needs theory emphasizes the importance of achieving goals or stimulating attachment to objects in augmenting human functions of employees. HAT can enhance the employees’ well-being by allowing them to interact with the colleagues and customers in ways that are not possible in reality.

*Visualizing unexpressed emotions*—Since the COVID-19 pandemic, it has become difficult to read others’ emotions due to physical distance and the need to communicate while wearing masks. Communication can be smoother when HATs are used to understand the emotions of others, and employees obtain job satisfaction. However, not many employees are willing to expose all their inner emotions even if the emotions can be easily expressed through automatic recognition. This may raise additional problems regarding which emotions should be expressed and which should be repressed. Pertaining to this topic, the following research questions need to be addressed: How can the accuracy of emotion detection be improved? How can the rules for emotional expression be determined? How can we avoid expressing emotions that we do not want to share with others?

*Highlighting feedback to performance*—Employees are reluctant to serve customers if they are not confident of their own skills [45,46]. Using HAT to highlight feedback on work performance, such as a praising voice for the employee or exaggerated visual effects, can satisfy achievement motivation and restore job satisfaction. However, if the highlight is extremely infantile and tedious, employees may become bored or even angry. It is also unlikely that a similar highlight will work for all employees [47]; thus, it is necessary to adjust feedback methods to suit the nature of the work. Pertaining to this topic, the following research questions need to be addressed: What highlights would be counterproductive? How can employee acclimation to feedback be avoided? How do effective feedback methods vary depending on the nature of the work?

### 4.7. Beyond Reality × Mastery Theory

The domain at the crosspoint of beyond reality and mastery theory emphasizes the importance of stimulating challenge and curiosity in augmenting the human functions of employees. As employees master HAT, they can enhance their well-being by performing tasks that they cannot perform in reality.

*Maturing augmented body*—Research on augmenting human bodies with machines is flourishing, and it is becoming possible to augment an actual body with, for example, a third arm, a sixth hand finger, and a tail [48]. The use of such body augmentation technologies requires employees to have different cognitive abilities than in conventional work. Therefore, employees need to adapt to technology that stimulates the challenge and promotes job satisfaction. However, using such body augmentation technologies in actual work is not the same as testing them in the laboratory. Employees usually work in company uniforms or in clothing that matches social norms, and since no company is likely to have uniforms suitable for employees with a third arm, companies need to provide clothing that naturally harmonizes with the workplace even when employees are wearing body augmentation technologies. In addition, body augmentation technologies will bring physical changes to the work environment. A third arm may not pass through a narrow door, and it is rare to find a workplace that provides a chair for a person with a tail to sit on. Pertaining to this topic, the following research questions need to be addressed: How can employees adapt to body augmentation? How can uniforms be designed to fit an augmented body? How should the workplace be redesigned to meet the needs of the augmented body?

*Applying others’ skills and knowledge*—Since the Stone Age, the scope of knowledge of individuals has become extremely narrow as tools have been developed. We cannot draw a bicycle correctly, nor can we clearly explain why the toilet flushes, even though we use it every day [49]. People exchange skills and knowledge in society [50], and the development of HAT enable the Internet of Abilities (IoA) [51] to leverage the capabilities of others that were not possessed by an individual employee, similar to how Trinity, the female protagonist in the movie *The Matrix*, can operate a helicopter in an instant based on a simple input. For example, by synchronizing the perspectives of skilled and novice employees, novice employees can utilize the knowledge from outside to enhance their own abilities [52]. Even connecting with non-humans may be possible in the future. However, we cannot yet synchronize with others as endlessly as the agents in *The Matrix*. We still need to learn to make the right decisions about when and what skills and knowledge to connect to, and it will take more time for our body and mind to be able to handle the burden of connecting to others. Pertaining to this topic, the following research questions need to be addressed: How does daily synchronization of perceptions strain the body and mind? How can the right skills and knowledge be selected for tasks? How can the skills and knowledge of non-humans be utilized?

### 4.8. Pursue Reality × Self-Determination Theory

The domain of the crosspoint of pursue reality and self-determination theory emphasizes the importance of arousing the employees’ intrinsic motivation while maintaining reality even when they use HAT. When introducing HAT, well-being can be enhanced by keeping the existing work environment as unchanged as possible or by reconstructing the real work environment in the virtual space, without harming the employees’ self-determination.

*Reconstructing realistic presence*—Employees experience heightened stress when the workplace changes as new technologies are introduced. To maintain the employees’ needs for autonomy and relationships while introducing HAT, the conventional workplace environment should be reconstructed including the presence of colleagues and customers, such that employees can perceive that they are in control of the work situation; this will restore their job satisfaction. It should be noted that the main goal is to make employees perceive that they are in control of the situation and not just replicate the superficial appearance of the workplace. We can recognize that the person we are talking to is really there in front of us even if we are talking remotely via a robot [53]. Reconstructing the authentic presence that goes beyond perceptual information is required to enhance the employees’ well-being. Pertaining to this topic, the following research questions need to be addressed: What should be reconstructed to make employees recognize the workplace as real? How can the presence of colleagues and customers be duplicated? How can authentic presence be achieved?

*Promoting embodiment in augmented telework*—Restoration of not only the surrounding environment but also the employee’s own sensory organs has a significant impact on their job satisfaction. Employees can work autonomously by being able to act with the same perception similar to that in conventional work, even in augmented telework such as tasks in virtual space or remote control [54]. In other words, it is essential to promote a sense of embodiment [55] in augmented telework, which allows employees to perceive a sense of competence in augmented telework that requires precise physical manipulation, such as in surgery. However, it also increases the risk of fatigue and discomfort when the body returns to the employee. The question is whether we can benefit only from positive effects while avoiding negative ones. Pertaining to this topic, the following research questions need to be addressed: How different is perception between conventional work and augmented telework? How do the effects of embodiment vary with the nature of the work? How can the negative effects of embodiment be overcome?

### 4.9. Pursue Reality × Needs Theory

The domain of the crosspoint of pursuit reality and needs theory highlights the importance of stimulating employee attachment to objects or achievement of goals while maintaining reality even when using HAT. Employees who essentially focus on needs theory emphasize extrinsic motivation from external factors that they cannot control. Therefore, identifying the factors in the surrounding environment that is the source of their job satisfaction and developing HAT that can maintain the functions of the environment can enhance the employees’ well-being.

*Supporting non-verbal feedback*—Communication is not limited to verbal communication. Employees also use facial expressions, eye gazes, and body language to communicate their intentions. By recognizing non-verbal feedback from others that is a factor that cannot be controlled through HAT, employees can regain their job satisfaction. The assistance provided by HAT enable employees with introverted personalities to communicate their intentions [56] and shorten psychological distance with others. However, non-verbal communication is more likely to be linked to power differentials within an organization than verbal communication [57]. A boss can convey certain instructions to his subordinate by simply coughing. Therefore, elimination of unnecessary power differences and facilitation of communication must be realized by HAT. Pertaining to this topic, the following research questions need to be addressed: How can non-verbal feedback be provided based on the employee’s personality? How can the relationship among employees be strengthened by supporting non-verbal feedback? How can power differences be overcome through non-verbal feedback?

*Pursuing reality in virtual space*—Future developments in HAT will lead to more work in virtual spaces. By reconstructing the same environment in the virtual space as in the real world, employees can obtain job satisfaction derived from external factors that cannot be controlled. This requires not only the reproduction of simple appearances of the workplace, such as the layout of physical properties, but also the reconstruction of the laws of nature, including the laws of physics. Indeed, a world in which everyone can fly in the sky is exciting, but employees will miss the natural order given by gravity and weather, when working in a virtual space as a simulation of reality. The love for houseplants is nurtured because they take time to grow, and that for work tools is also because their history is engraved in the rust. However, no matter how desirable it is to reconstruct the laws of nature, no one would want to encounter a natural disaster even in a virtual space. Pertaining to this topic, the following research questions need to be addressed: What are the natural laws that have been discarded in the process of reconstructing virtual space? What are the difficulties in reconstructing the laws of nature? What are the natural laws that decrease the employees’ job satisfaction?

### 4.10. Pursue Reality × Mastery Theory

The domain at the crosspoint of pursue reality and mastery theory emphasizes the importance of stimulating the employees’ challenge and curiosity while maintaining reality when they use HAT. The introduction of HAT often requires employees to move in different manners from conventional physical movements. For example, virtual reality technologies require employees to perform tasks through cognitive functions without much physical movement. However, the mastery of a task is not necessarily a function of cognition but also of physical function. Mastering the physical capabilities of employees similar to those in reality can promote job satisfaction and enhance well-being.

*Accompanying physical movements*—The employees’ bodies should be stimulated synchronously with augmented telework. Employees who are initially confused by the synchronization of augmented telework with their real bodies will gradually master the technology. This development process is challenging for employees and encourages job satisfaction. In addition, the mastery of physical movements accumulates tacit knowledge [58] in employees that can increase their value in the labor market. In contrast, the accompanying physical movements impose an actual physical burden on the employees that makes their own health management even more important. Pertaining to this topic, the following research questions need to be addressed: How can physical movement be synchronized with augmented telework? How can employees acquire tacit knowledge? How can the safety of employees be ensured?

*Fulfilling potential capabilities*—As new technologies are introduced in society, we degrade our existing physical and cognitive abilities to adapt to them [59]. In contrast, employees can enhance their well-being by fulfilling their own potential capabilities through HAT. They will be challenged and motivated to work harder as their strengths are further developed, and the range of tasks they can accomplish is expanded. Furthermore, if employees can gradually develop their potential capabilities in their daily work, they might positively impact their own lives [60]. There is no strong reason to believe that creativity developed through work cannot be useful in daily life. It is no exaggeration to say that whether we can fulfill potential capabilities in the short term depends on how we behave in our daily work, as people nowadays place work at the center of their lives. Pertaining to this topic, the following research questions need to be addressed: How can the potential capabilities needed for a job be assessed? How can employees realize their potential capabilities? How can employees be encouraged to sustain the potential capabilities they developed in their daily life?

## 5. Conclusions

The COVID-19 pandemic has prompted the establishment of new work styles. The development of HAT is urgently required, but there has not been any sufficient agenda for the R&D of technologies to enhance employees’ well-being. Therefore, GTA was used to clarify the technology demand and job satisfaction of employees during the pandemic and proposed an R&D agenda based on the crosspoint of those two factors. The development of HAT according to the proposed framework and the agenda can support the improvement of employees’ well-being through technology in the new normal.

The theoretical implication of this study is the development of two models: a technology demand model and a job satisfaction model for HAT in the COVID-19 pandemic. The demand model revealed that the demand for HAT by employees during the pandemic can be explained in terms of the two axes of reality and accessibility. While previous studies attempting to identify demands of HAT users only discussed a single axis of reality or accessibility [8,61], our model explains how these two axes can describe technology demands about HAT comprehensively. Furthermore, the job satisfaction model revealed that the job satisfaction of employees during the pandemic can be explained based on the two axes of the locus of control and intrinsic motivation. The findings that the locus of control and intrinsic motivation form job satisfaction through technology use are consistent with previous studies [62,63]. In addition, this study mapped how three motivation theories (i.e., self-determination theory, needs theory and mastery theory) can expound job satisfaction. The two models provided new insights into the research on technology and the employees’ well-being.

The practical implication is that this study presented topics for R&D that should be promoted for improving the employees’ well-being. Although the two models presented in this study can comprehensively explain technology demand and job satisfaction, they are somewhat abstract at an individual level and difficult to apply in practice. Therefore, this study integrated the two models and proposed a comprehensive framework for the R&D of HAT based on the crosspoint of the two models. This framework can be used for developing HAT to support job performance while addressing the impact on the employees’ well-being. Previous studies providing R&D agenda for HAT generally focus on specific topics or tools such as performance of military [64], brain-computer interface [65], and retail business [66]. This study describes comprehensive employee well-being from the viewpoints of technology demand and job satisfaction. The nine domains of the crosspoint framework will provide insights for R&D of new HAT.

Although this study proposed a framework for understanding both technology demand and employee well-being, the perspectives emphasized in technology development vary considerably for each industry. The limitation of this study is that the perspectives of technology development required by each industry have not been analyzed. Future study will focus on clarifying the R&D agenda for each industry.

## Figures and Tables

**Figure 1 ijerph-19-01195-f001:**
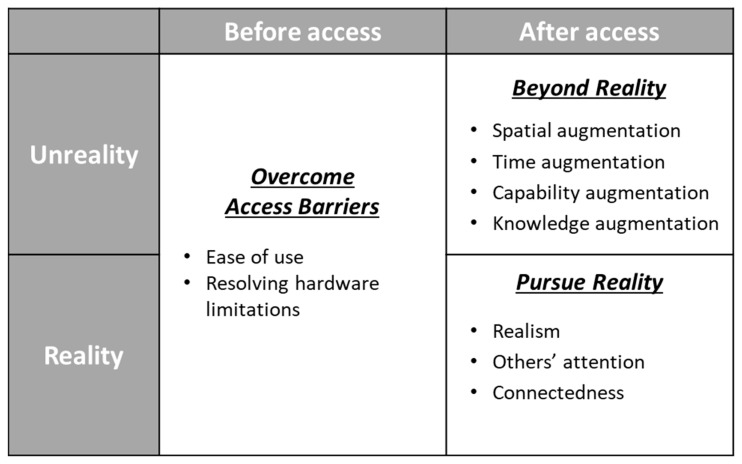
Model of demand for HAT in the COVID-19 pandemic.

**Figure 2 ijerph-19-01195-f002:**
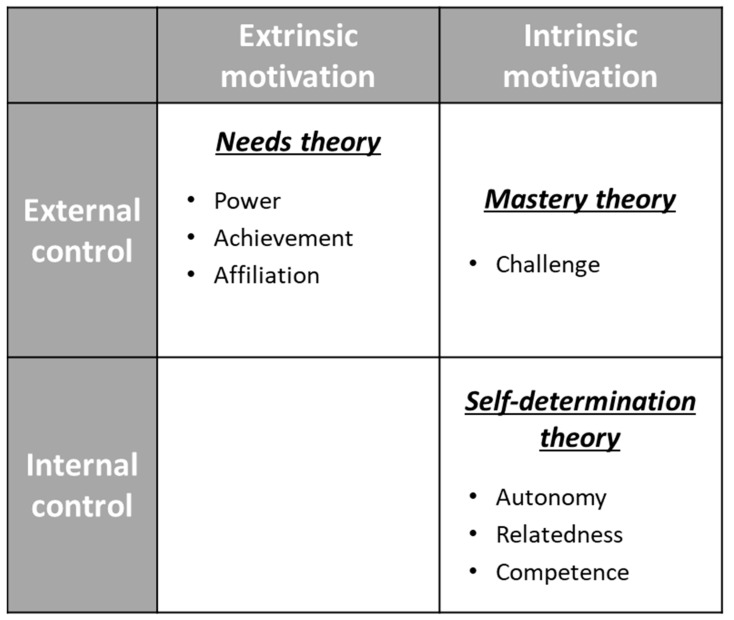
Model of job satisfaction in the COVID-19 pandemic.

**Table 1 ijerph-19-01195-t001:** Profile of respondents.

No.	Sex	Age	Occupation	Overview of Life Changes after the COVID-19 Pandemic
1	M	30s	Data analyst	He moved to a rural area after the pandemic and began to switch to teleworking. With no more commuting, he regained a human-like life.
2	M	40–50s	Engineer/ Manager	He generally goes to the office even after the pandemic. He feels that there have been positive changes in the format of meetings and sales, such as cost reduction.
3	F	40–50s	Manager	She is a researcher and also manages a group home for the elderly and teaches piano to children. The group home, where employees interact with the elderly, has had particular difficulty in dealing with the COVID-19. Although she continues to teach piano face-to-face, the proportion of teleworking has increased in the work of the researcher and manager.
4	M	30s	Tour organizer	He works as an organizer of study tours. He switched to teleworking and had more time to spend with his children after the pandemic, but there were both good and bad aspects to this.
5	M	20–30s	Management consultant	She started own business and moved to a rural area after the pandemic. Her clients are from existing relationships that developed when she was working in an urban area. She is teleworking.
6	M	20–30s	Tour guide	He is a tour guide in sole proprietorship. He used to offer self-made tours to overseas clients face-to-face, but switched to offering online tours after the pandemic.
7	M	30–40s	Office worker	He teleworked immediately after the pandemic. He could do housework in spare time while teleworking. Subsequently, he started going to office every day as usual. He feels that the pandemic has not changed life and work much.
8	F	30–40s	Office worker	Since the pandemic, she has mainly been teleworking that has reduced her stress as she can spend the lost commuting time on housework.
9	F	40–50s	Office worker	She was transferred during the pandemic, and she started teleworking. Teleworking eliminated the 4 h commute and allowed her to spend more time sleeping and exercising instead.
10	F	30s	High school teacher	She started going to a training gym and spending more time studying about the class content after the pandemic. She has started to create online class content, but mostly teaches face-to-face.
11	M	20–30s	High school teacher	He has been devoting much time to coaching club activities, but his time for this has considerably reduced. He feels that there are both good and bad sides to this change though he has more free time.
12	M	50–60s	High school teacher	While teaching information and mathematics, he also coordinates student counseling and communication among the faculty. He teaches face-to-face mainly. He faces difficulty in communicating, as everyone wears masks since the pandemic.
13	F	50s	Researcher	She teleworks about one day a week and her life is not much different from that before the pandemic. Online meetings have become more frequent, and she feels that her working hours have increased after the pandemic.
14	M	30–40s	Researcher	He generally goes to office after the pandemic. He feels that he communicates more with people to see their faces, as online meetings have become more prevalent than email communications.
15	M	30s	Researcher	He lives alone and is happy that he can telework more after the pandemic, with more time to do housework. However, he still goes to office for 80% of his working hours.
16	M	40s	Researcher	He used to go out for meetings with people outside the company before the pandemic. He started to telework and moved to a rural area after the pandemic.
17	F	30s	Researcher	She took maternity leave before the pandemic and returned in autumn 2020. She mainly works by teleworking since she resumed work, but she feels that the change in life and work style is not only due to the pandemic, but also due to childcare.
18	M	40s	Researcher	He used to commute 3 h every day, but teleworked more than half of the time after the pandemic. He feels online meetings make conversations more aggressive than face-to-face ones.
19	M	20–30s	Gardener	There was no significant change in work style after the pandemic. Demand for houseplants increased as people went out less.
20	F	50–60s	Cooking teacher	All the face-to-face cooking classes were now being conducted online. She is having trouble using video call tools because many of her clients are elderly. She began to live a slower life after the pandemic, going for walks in a forest every day.

**Table 2 ijerph-19-01195-t002:** Crosspoint framework for R&D Agenda of HAT.

	Overcome Access Barriers	Beyond Reality	Pursue Reality
**Self-Determination Theory**	Improving interface usability Improving operation ability by perceptual stimulation	Visualizing potential ability and conditions Moderating disparities in augmented capabilities	Reconstructing realistic presence Promoting embodiment in augmented telework
**Needs Theory**	Imparting personality to instructional systems Building an online community	Visualizing unexpressed emotions Highlighting feedback to performance	Supporting non-verbal feedback Pursuing reality in virtual space
**Mastery Theory**	Stepwise unlocking of technological functions Restoring physical and cognitive functions and supporting working	Maturing augmented body Applying others’ skills and knowledge	Accompanying physical movements Fulfilling potential capabilities

## Data Availability

The data presented in this study are available on request from the corresponding author. The data are not publicly available due to restrictions of privacy.

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
