# Peer review of "Human Augmentation Technologies for Employee Well-Being: A Research and Development Agenda"

_ijerph, 2022, doi:10.3390/ijerph19031195_

Round 1
Reviewer 1 Report
- I think that the overall structure and writing of the introduction part are not clear and well-aligned. Please clearly describe those things. As you already know, the introduction section is one of the most important parts to draw readers' attention and provide guidelines for them to facilitate a clear understanding of the paper.
- The main research questions are not well developed.
- Mention student contribution in introduction section.
- The theoretical assumptions of the research questions development are confusing and are not well justified.
- The discussion section is missing.
-
-Although the authors have attempted to explain the contributions and implications of the paper, I think that the overall quality of the explanations is low. Please provide more elaborated explanations to demonstrate its theoretical and practical contributions.
Author Response
Reviewer 1
- I think that the overall structure and writing of the introduction part are not clear and well-aligned. Please clearly describe those things. As you already know, the introduction section is one of the most important parts to draw readers' attention and provide guidelines for them to facilitate a clear understanding of the paper.
- Thank you for your comment. We have accordingly revised the Introduction section (lines 41–73) to improve the quality of our paper.
- The main research questions are not well developed. The theoretical assumptions of the research questions development are confusing and are not well justified.
- We thank your insightful comment. We have accordingly improved the process of RQ development (lines 58–65). The two RQs are set up from the perspective of the balance model of employee well-being.
- Mention student contribution in introduction section.
- We had conducted the study without support from students.
- The discussion section is missing.
- We wrote the discussion section and provided the crosspoint framework in Section 4 (lines 419–736). Following this, we have argued the contributions of our study in the Conclusion section, to distribute content more evenly and avoid any lengthy sections.
- -Although the authors have attempted to explain the contributions and implications of the paper, I think that the overall quality of the explanations is low. Please provide more elaborated explanations to demonstrate its theoretical and practical contributions.
- Thank you for your insightful comment. We have accordingly added explanations for the theoretical and practical contributions (lines 749–771).

Reviewer 2 Report
The article is very interesting. The study sample presented (pilot) very important and certainly the formulation of some framework for further research is now much needed. Is the number of respondents of 20 not too small, even assuming a pilot study, and could the choice of method, in this case " snowball sampling" in the selection of respondents, be more extensively, strongly explained/argued.
My feeling is that the number of keywords should be rethought and reduced. I would think about improving, first of all, the graphic design of the article, at the moment the form is not 100% clear to me - maybe add a diagram, a chart of proceedings also in addition to tables, etc.
I miss information in the article about similar studies being conducted and commenting on their results or if they are being conducted - obviously in the context of the Covid-19 pandemic.
Author Response
Reviewer 2
- The article is very interesting. The study sample presented (pilot) very important and certainly the formulation of some framework for further research is now much needed. Is the number of respondents of 20 not too small, even assuming a pilot study, and could the choice of method, in this case " snowball sampling" in the selection of respondents, be more extensively, strongly explained/argued.
- We thank you for your insightful comment. To improve the explanation, we have added the sentence, “We also considered the balance of gender, age and occupation when recruiting respondents.” (lines 97–98).
- My feeling is that the number of keywords should be rethought and reduced. I would think about improving, first of all, the graphic design of the article, at the moment the form is not 100% clear to me - maybe add a diagram, a chart of proceedings also in addition to tables, etc.
- Thank you for kindly comment. We have accordingly updated the keywords. To improve the clarity of the manuscript, we have revised the explanations in the Introduction section (lines 41–73). In addition, we have improved the design of Figure 1 and 2.
- I miss information in the article about similar studies being conducted and commenting on their results or if they are being conducted - obviously in the context of the Covid-19 pandemic.
- Thank you for your insightful comment. We have added explanations in the Introduction (lines 41–44) and Conclusion (lines 738–771) sections.

Reviewer 3 Report
I hope authors find interesting my suggestions to improve the manuscript:
Thanks for the opportunity to review this interesting work. In order to improve it, I would like to make some considerations to the authors.
TITLE: I am not sure the title: ”Human Augmentation Technologies for Employee Well-Being: A Research and Development Agenda” represents well this work because the word “agenda” is unclear to me in that context. Sorry if it is for my English skills or transcultural translation. I suggest something like “Human Augmentation Technologies and Employee Well-Being during COVID: A Research from workers view ” or whatever similarly.
INTRODUCTION: Could you introduce a brief background according to the socio-economic factors, the technological knowledge and teleworking rate before COVID of the general population of the geographic sample? I think it could be relevant in order to understand the results of the work.
Too many Keywords. Could you consider to delete some of these words: research and development; employee; technology or work? Or can you introduce most specific others like work style or technologies for work support?
About research and development (R&D), I think (RD) is more suitable with journal style.
DISCUSSION:
Line 464. Is this comparison necessary? “similar to how Theodore fell in love with Samantha when he recognized her personality in the movie “Her””.
Line 602 “similar to how Trinity, the female protagonist in the movie “The Matrix,” can operate a helicopter in an instant based on a simple input
I do not like movies references at all.
I think Discussion section is right but excessive bibliographic references are used (e.g. 45,46 or 49) In my opinion are useless references.
REFERENCES:
Self citation?:
- Ho, B.Q.; Shirahada, K. Barriers to Elderly Consumers’ Use of Support Services: Community Support in Japan’s Super-Aged 786 Society. J. Nonprofit Public Sect. Mark. 2020, 32, 242–263.
- Ho, B.Q.; Shirahada, K. Analysis of the Characteristics of Local Elderly’s Buying Behavior. J. Japan Assoc. Reg. Dev. Vital. 2015, 6, 814 71–78.
- Ho, B.Q.; Shirahada, K. Actor Transformation in Service: A Process Model for Vulnerable Consumers. J. Serv. Theory Pract. 2021, 837 31, 534–562.
- Ijuin, K.; Ogata, K.; Watanabe, K.; Miwa, H.; Yamamoto, Y. Proposing Remote Video Conversation System “PARAPPA”: Deliv-865 ering the Gesture and Body Posture with Rotary Screen. In Proceedings of the 2021 30th IEEE International Conference on Robot 866 & Human Interactive Communication (RO-MAN), Vancouver, BC, Canada, 8-12 August 2021.
.
Author Response
Reviewer 3
- TITLE: I am not sure the title: ”Human Augmentation Technologies for Employee Well-Being: A Research and Development Agenda” represents well this work because the word “agenda” is unclear to me in that context. Sorry if it is for my English skills or transcultural translation. I suggest something like “Human Augmentation Technologies and Employee Well-Being during COVID: A Research from workers view ” or whatever similarly.
- Thank you for your insightful comment. As you have pointed out, the word “agenda” may be somewhat vague. However, we can find numerous articles with “research agenda” in the title in several research fields. In this study, we provided an agenda for both researchers and developers of human augmentation technologies by proposing a crosspoint framework. Therefore, we believe that the original title is applicable in this context. Although the suggestion “workers view” describes the contents of the paper well, we believe that the original title is a better fit for our main target readers.
- INTRODUCTION: Could you introduce a brief background according to the socio-economic factors, the technological knowledge and teleworking rate before COVID of the general population of the geographic sample? I think it could be relevant in order to understand the results of the work.
- I would like to thank you for giving us concrete suggestions. We have added the statistics of teleworking in the Introduction section (lines 41–44) of the revised manuscript.
- Too many Keywords. Could you consider to delete some of these words: research and development; employee; technology or work? Or can you introduce most specific others like work style or technologies for work support?
- Thank you for your insightful comments. We deleted “technology” and “work” and added more specific keywords in lieu of these (i.e. “teleworking” and “virtual reality”).
- About research and development (R&D), I think (RD) is more suitable with journal style.
- Thank you for kind suggestion. However, we believe “R&D” is better than “RD,” because “R&D” is jargon and used in many research fields.
- I do not like movies references at all. I think Discussion section is right but excessive bibliographic references are used (e.g. 45,46 or 49) In my opinion are useless references.
- I understand the reviewer’s concerns. While movie references tend to be avoided in academic writing, we believe that showing concrete examples helps readers to imagine the kind of technologies discussed in the paper and enables us to provide concrete examples with short sentences.
- Self citation?:
- We referred to the articles that were suitable for the descriptions. Therefore, we believe that these cases are the appropriate citations.
